# Molecular cytogenetic mapping of *Secale strictum* introgressions in a perennial tetraploid rye and its diploid progenitor

Ahmed Ali Hamad[1,2], Diána Makai[1], László Sági[1], Ákos Tarnawa[2], Adél Sepsi [1]*, Dávid Polgári[1,2]*

**1** Department of Biotechnology and Reproduction Biology, HUN-REN Centre for Agricultural Research, Martonvásár, Hungary, **2** Institute of Agronomy, Hungarian University of Agriculture and Life Science, Gödöllő, Hungary

* sepsi.adel@hun-ren.atk.hu (AS); polgari.david@uni-mate.hu (DP)

## Abstract

Perennial rye offers a means to reduce soil disturbance in water-limited cropping systems; however, the use of interspecific germplasm is often constrained by variable fertility and structurally heterogeneous genomes that are difficult to monitor during selection. Here, we selected a new perennial rye line, 'Keleti1', from a *Secale cereale* × *S. strictum* hybrid population and generated a tetraploid derivative, 'Keleti1T', by colchicine-induced genome duplication. Comparative phenotyping of matched diploid and tetraploid plants showed that genome duplication was associated with reduced plant height, increased thousand-grain weight and fertile tiller number per plant. In contrast, per-spike reproductive performance and seed number per plant exhibited substantial inter-individual variation in tetraploids. Flow cytometry and cytological analyses were used to confirm tetraploid status prior to molecular cytogenetic characterisation. To enable chromosome-level tracing of parental contributions, we applied fluorescence *in situ* hybridisation using the subtelomeric repeat pSc119.2 and 5S rDNA. These markers revealed polymorphisms on chromosomes 1R and 6R, as well as on 3RL, 4RL, 7RL and 5RS, allowing *S. strictum* chromatin segments to be distinguished in both diploid and tetraploid backgrounds. Heteromorphic chromosome pairs were detected in all plants analysed, consistent with the outcrossing nature of rye, and non-canonical hybridisation patterns indicated frequent structural rearrangements. Together, these results provide cytogenetic markers for monitoring introgression and document how genome duplication alters agronomic and reproductive traits in a perennial interspecific rye derivative.

## Introduction

Perennial rye, derived from *Secale strictum* (C. Presl) C. Presl and its hybrids with cultivated rye, is primarily used for pasture grazing and green manure and has

**Data availability statement:** All relevant data are within the manuscript and its Supporting Information files.

**Funding:** The research was supported by the Hungarian Research, Development and Innovation Office (NKFIH, TKP2021-NKTA) (AS), the Hungarian National Laboratories Program (grant number RRF-2.3.1-21-2022-00007) (LS) and the Doctoral School of Plant Science of the Hungarian University of Agriculture and Life Science (AAH). The funders had no role in study design, data collection and analysis, decision to publish, or preparation of the manuscript.

**Competing interests:** The authors have declared that no competing interests exist.

attracted renewed interest as a component of low-input and water-limited cropping systems. Its perennial growth habit provides persistent ground cover, reduces soil disturbance, and supports extensive root systems that improve soil structure and resource retention [1,2]. In contrast, annual rye (*Secale cereale* L.; 2n = 2x = 14, ($R^cR^c$ genome) is a major cereal crop cultivated for grain, forage, and cover cropping, valued for its tolerance to cold, drought, and poor soils, as well as its rapid establishment and reliable seed production under marginal conditions [3,4]. Integrating perennial growth habit into cultivated rye has therefore been considered a strategy to combine the agronomic robustness and seed productivity of annual rye with the soil-conserving benefits of perennial systems.

Within the genus *Secale*, mountain rye (*Secale strictum* (C. Presl) C. Presl) (formerly *S. montanum* Guss.) (2n = 2x = 14, $R^sR^s$) is the only naturally perennial species and represents a key genetic resource for introducing perennial growth habit into cultivated rye [5,6]. Analyses of chloroplast and mitochondrial non-coding regions indicate a close genetic relationship between cultivated rye and wild *Secale* taxa, including *S. strictum*, consistent with historical gene flow and supporting their use in interspecific hybridisation programmes [7]. Interspecific hybrids between these species have been produced repeatedly and can express perenniality, but their agronomic performance is frequently compromised by reduced fertility and irregular chromosome behaviour during meiosis [8,9]. These defects have been linked to structural differences between the parental genomes, including translocations involving several rye chromosomes, which promote multivalent formation, genome instability, and aberrant segregation [10–12]. As a consequence, the retention and transmission of introgressed chromatin are highly variable, complicating selection and breeding.

Whole-genome duplication has been proposed as a means to mitigate some of these cytogenetic constraints and improve grain yield in interspecific rye hybrids. Genome duplication can increase opportunities for homologous chromosome pairing and, in some cases, reduce multivalent formation relative to diploid hybrids carrying structurally divergent chromosome sets [13–15]. At the same time, polyploidisation is known to induce extensive genome restructuring, including sequence rearrangements and changes in repetitive DNA composition, particularly in interspecific polyploids [16], in addition to altering plant architecture and reproductive traits [15–17].

Polyploidisation alters plant architecture and reproductive traits, often increasing seed size and biomass while affecting fertility [17–19]. In rye, however, the effects of genome duplication in perennial interspecific backgrounds remain insufficiently characterised, and it is unclear how polyploidisation interacts with the extensive structural heterozygosity typical of outcrossing *Secale* species.

A further limitation in perennial rye improvement is the difficulty of tracing parental chromosome segments and structural variation during selection. Classical cytogenetic approaches, including C-banding, enabled early karyotype analyses in rye but lack the resolution required for reliable interspecific comparisons and for identifying introgressed chromatin [20–22]. Fluorescence *in situ* hybridisation (FISH) using well-characterised repetitive DNA probes provides a robust framework for chromosome identification in rye and enables chromosome-arm-level discrimination between

parental genomes [23–25]. Despite its demonstrated utility, systematic application of FISH to perennial rye lines and their polyploid derivatives has remained limited.

In this study, we address these constraints using a newly selected perennial rye line, 'Keleti1', derived from a *Secale cereale* × *S. strictum* hybrid population [26], and its colchicine-induced tetraploid derivative, 'Keleti1T'. Specifically, we aimed to: (i) generate and phenotypically characterise a tetraploid derivative of 'Keleti1' to quantify trait shifts associated with genome duplication; (ii) use pSc119.2 and 5S rDNA FISH markers to define parental chromosome segments in both diploid and tetraploid backgrounds; and (iii) assess the extent of chromosome heteromorphism and structural variation within and between ploidy levels. By integrating phenotypic analysis with chromosome-level cytogenetic mapping, this work establishes a framework for monitoring introgression and genome stability in perennial interspecific rye derivatives.

## Materials and methods

### Plant material

The perennial rye variant 'Keleti1' was selected from a heterogeneous backcross population of the interspecific hybrid *Secale cereanum* ($2n = 2x = 14$, $R^cR^s$). The initial plant was clonally propagated, and the progenies were grown under field conditions in a single plot containing 100 individuals. Seeds collected from open-pollinated somaclonal sister spikes were subjected to genome duplication. Since *Secale cereanum* originated from a cross between the cultivated winter rye variety 'Kisvárdai' (*Secale cereale*, $2n = 2x = 14$, $R^cR^c$) and the perennial mountain rye (*Secale strictum*, $2n = 2x = 14$, $R^sR^s$), both parental genotypes were used for cytological comparisons in this study.

### Colchicine treatment

For synchronized germination, fungicide (Lamardor 400 FS, Bayer Crop Science, Monheim am Rhein, Germany) coated 'Keleti1' seeds were immersed in a 600 µM gibberellic acid solution (G0907 Gibberellic Acid $A_3$, Duchefa Biochemie, Haarlem, The Netherlands) for 24 hours at 4°C in the dark. Seedlings with a coleoptile length of 5–8 mm were kept on moist filter paper in Petri dishes at 4°C until further treatment. To induce whole-genome duplication, 100 selected seedlings were immersed in 100 mL of 0.5% colchicine solution containing 1.5% dimethyl sulfoxide (DMSO, Sigma-Aldrich, St. Louis, MO, USA) for 90 minutes at room temperature. The treated seedlings were rinsed five times with sterile water to prevent further colchicine effects and were then transferred to Petri dishes lined with filter paper saturated with a 600 µM gibberellic acid (GA) solution. Regeneration occurred in a Pol-Eko ST 700 growth chamber (Pol-Eko Aparatura sp. j., Wodzisław Śląski, Poland) at 18°C with a 12/12-hour day/night cycle. Regenerating seedlings were transferred to Jiffy peat pellets (Ø 33 mm) (Jiffy-7, 33 mm, Jiffy International AS, Kristiansand, Norway).

### Flow cytometry

The ploidy of the $C_0$ plants (colchicine-treated individuals) and their $C_1$ progeny was determined at three different developmental stages (5–6 leaves, tillering, and flowering) using flow cytometry with the CyStain UV Precise P kit (Sysmex Partec GmbH, Görlitz, Germany). Fresh leaf segments (1 cm²) were chopped with a razor blade in a plastic Petri dish (50 mm in diameter) containing 500 µL of Nuclei Extracting Buffer (Sysmex Partec GmbH). After a one-minute incubation, the liquid fraction was transferred to a 50 µm CellTrics filter (Sysmex Partec GmbH) and washed into the sample tube with 2 mL of Staining Buffer (Sysmex Partec GmbH). The DNA content of the nuclei was measured after a five-minute incubation using a Sysmex CyFlow Space flow cytometer (Sysmex Partec GmbH) equipped with UV LED illumination. Relative genome size was determined using Sysmex Partec Flomax 2.11 software. Ploidy was assessed by comparing the relative genome size to untreated (diploid) control plant samples.

## Fertility and phenotypic assessment

Seed set per spike was calculated as the average number of grains per spike for each plant. The number of spikelets was recorded for each spike, and fertility was determined as the number of grains per spikelet. The number of fertile tillers was defined as the total number of spikes produced per plant during a single reproductive season (June-July), encompassing both the early harvest followed by a late harvest.

## Chromosome preparation for cytology

Freshly emerged root tips of approximately two cm were collected directly from the diploid and tetraploid plants and washed with distilled water. Collected roots were transferred into 10 mL glass vials filled with ice-cold Milli-Q water containing approx. 1/3 volume of melting ice. Vials were placed in a Thermocool box filled with ice and incubated in a cold room for at least 26 h. The roots were then placed in freshly prepared Clarke's fixative, and incubated at 37°C for 5 days. Fixed roots were submerged in 1% acetocarmine at room temperature (RT) for 2 h and stored in fresh Clarke's fixative at −20°C for 14 days. Fixed root tips were immersed in a 45% acetic acid solution for 10 minutes and placed on a microscope slide. The root-tip was cut and discarded and cells were squeezed out into a drop of 45% acetic acid. After placing the coverslip, the slides were heated for a few seconds over a spirit burner. The coverslips were removed after the slides were frozen in liquid nitrogen. The slides were air-dried overnight and stored at −20°C until further use.

## Preparation of the FISH probes

Fluorescence *in situ* hybridization was carried out by PCR amplification of the rye subtelomeric heterochromatic sequence pSc119.2 [27,28] and the 5S rDNA sequences [29–31] followed by labelling with AF594 (AF594 NT Labelling Kit, PP-305 L-AF594, Jena Bioscience, Germany) and AF488 (AF488 NT Labelling Kit, PP-305 L-AF488).

## Fluorescence *in situ* hybridisation

*In situ* hybridization was performed as described by [32]. In brief, the chromosome preparations were digested with a 50 mg/mL pepsin-1 mM HCl solution for 1–3 min (depending on the amount of cytoplasm observed) at 37°C which was followed by a post-fixation in 4% (w/v) PFA (diluted from 16% stock, 28,908, Thermo Scientific) for 10 min (RT). The final hybridisation mixture consisted of 60% (v/v) deionised formamide (F9037, Sigma-Aldrich), 10% (w/v) dextran sulphate (D8906, Sigma-Aldrich), and 2X Saline Sodium Citrate buffer. Seventeen µL of the hybridisation mixture was completed with 40–50 ng of each of the labelled probes, denatured at 85°C for 8 min 30 sec and placed on ice for five minutes. Slides containing the probe mixture were additionally denatured at 75°C for 3 min 30 sec. Hybridisation was performed at 37°C overnight. After post-hybridisation washings, the slides were covered with 24 × 32 mm coverslips in 18 µL of Vectashield antifade solution with DAPI (H1200, Vector Laboratories, Burlingame, CA, USA). Images were taken by an SP8 confocal laser scanning microscope (Leica Microsystems GmbH, Wetzlar, Germany) equipped with an HC PL APO CS2 63×/1.40 oil immersion objective.

## Perennial growth habit

The perennial growth habit of the diploid 'Keleti1' base material was tested in an outdoor field experiment consisting of four plots, each containing 50 plants. Seeds were germinated on moistened filter paper in Petri dishes and the germinating plantlets were transferred to Jiffy peat pellets (Ø 33 mm) (Jiffy-7, Jiffy International AS, Kristiansand, Norway). After vernalization at 4°C under an 8 h light/ 16 h dark photoperiod for 8 weeks, the plants were transplanted to the field in April. The experimental plots were covered with agrotextile sheets perforated with 5 × 5 cm holes to prevent volunteer plants from emerging and thereby avoid bias in the evaluation. Regrowth percentage was determined in the following spring by manually recording the number of regrowing/dead plants.

To assess the maintenance of the perennial growth habit at the tetraploid level, 14 diploid and 14 tetraploid plantlets were grown in parallel under semi-outdoor conditions. Germination and vernalization were carried out in the same way as in the outdoor field experiment. The plants were cultivated in 5 L pots filled with soil and placed outdoors. To prevent desiccation, pots were manually irrigated whenever soil drying was observed. Regrowth percentage was evaluated in the second growing season by manually recording the number of regrowing/dead plants.

### Statistical analyses

Statistical analyses were conducted using the open-source statistical software JASP 0.16 (University of Amsterdam, Amsterdam, the Netherlands). Descriptive statistics, including mean and standard deviation, were calculated for all measured variables. The normality of data distribution was assessed using the Shapiro-Wilk test, while homogeneity of variances was evaluated using Levene's test. For comparisons between groups, an independent Student's t-test was applied when normality assumptions were met. In cases where normality was violated, the non-parametric Mann-Whitney U test was used. Where the group variances were unequal violation was corrected by using an adjusted t-statistic based on the Welch method. All statistical tests were performed with a significance level set at $p < 0.05$.

## Results

### Generation and validation of diploid and tetraploid 'Keleti1' lines

A perennial rye line, 'Keleti1' ($2n = 2x = 14$), was selected from a heterogeneous backcross population derived from *Secale cereale* × *S. strictum*. To assess the effect of polyploidy, whole-genome duplication was induced by colchicine treatment of seeds obtained from open-pollinated, clonally propagated 'Keleti1' plants (Fig 1A). Of the 100 treated seedlings, 91 regenerated successfully. Flow cytometry analysis of $C_0$ plants revealed three ploidy classes based on nuclear DNA content: 66 diploids (72.5%), seven tetraploids (7.7%), and 18 mixoploids (19.8%) (Fig 1B; S1 Fig). Chromosome counting in root-tip cells of $C_1$ progeny confirmed somatic chromosome numbers of $2n = 14$ for diploids and $2n = 28$ for tetraploids (Fig 1C). Only stable tetraploid plants were retained for further analyses.

### Effects of genome duplication on plant architecture and yield components

Comparative phenotyping was conducted on matched diploid and tetraploid plants (n = 14 per cytotype; Fig 2A; S2 Fig). Genome duplication was associated with consistent changes in seed morphology. Tetraploid plants produced significantly larger seeds than diploids, as reflected by a higher thousand-grain weight (mean 18.949 ± 1.391 g in tetraploids vs. 15.758 ± 1.807 g in diploids; Student's t-test, $p < 0.001$; Fig 2B, C) and by significant increases in both seed length and seed width (Student's t-tests, $p < 0.001$; S3 Fig; S1 Table).

Genome duplication also affected plant architecture (Fig 2A, D). Tetraploid plants were significantly shorter than diploids (mean 120.714 ± 5.797 cm vs. 134.857 ± 5.005 cm; Student's t-test, $p < 0.001$; Fig 2A; S3 Fig), but produced a significantly higher number of fertile tillers (spikes) per plant (mean 24.857 ± 8.047 in tetraploids vs. 17.500 ± 7.014 in diploids; Student's t-test, $p = 0.016$; Fig 2A, G; S2 Fig; S1 Table). Reproductive efficiency at the spike level showed substantial inter-individual variation with mean seed set per floret being lower and more variable than in diploids (mean 0.271 ± 0.154 vs. 0.327 ± 0.076; Mann–Whitney U test, $p < 0.001$; Fig 2E). Tetraploid spikes also varied in architecture, with fewer florets per spike on average (mean 82.39 ± 12.06 vs. 107.11 ± 15.91; Mann–Whitney U test, $p < 0.001$; S3 Fig) and a lower number of seeds per spike (mean 22.046 ± 13.179 vs. 34.404 ± 6.789; Mann–Whitney U test, $p < 0.001$; Fig 2F; S1 Table). At the whole-plant level, seed number varied widely, with several tetraploid individuals reaching moderate to high values within the observed range (Fig 2H).

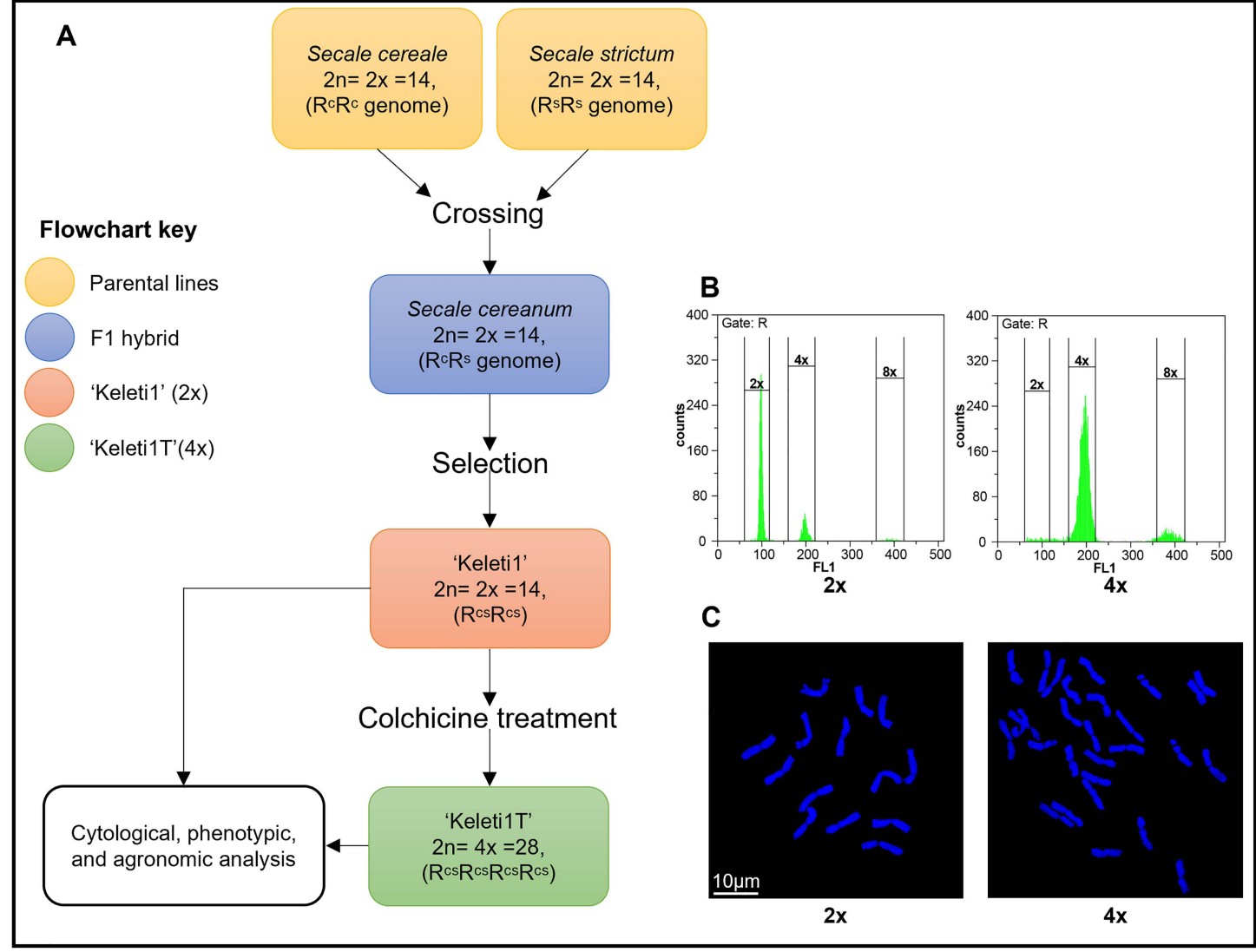

**Fig 1. Induction and confirmation of tetraploidy in perennial rye. (A)** Schematic workflow illustrating the generation of perennial rye from a *Secale cereale* × *S. strictum* cross, selection of the diploid perennial line ('Keleti1', 2x), colchicine-induced chromosome doubling, and subsequent analyses of the tetraploid derivative ('Keleti1T', 4x). Colours indicate parental lines, F₁ hybrid, diploid 'Keleti1', and tetraploid 'Keleti1T', as shown in the key. **(B)** Flow cytometry histograms showing relative nuclear DNA content of diploid (2x) and tetraploid (4x) plants. **(C)** Representative chromosome spreads of diploid ('Keleti1', 2x) and tetraploid ('Keleti1T', 4x) plants counterstained with DAPI. Scale bar = 10 μm.

## Maintenance of perennial growth habit at diploid and tetraploid levels

The perennial growth habit of diploid 'Keleti1' was evaluated under field conditions across four replicated plots (50 plants per plot). In the subsequent growing season, regrowth rates ranged from 76% to 96% among plots, resulting in an overall regrowth rate of 86% (Fig 3). Perenniality at the tetraploid level was assessed in a semi-outdoor pot experiment using clonally propagated diploid and tetraploid plants (n = 14 per cytotype). Following spike harvest and removal of aerial bio-mass, all plants regenerated successfully in the following season, indicating maintenance of the perennial growth habit in both diploid and tetraploid backgrounds (S4 Fig).

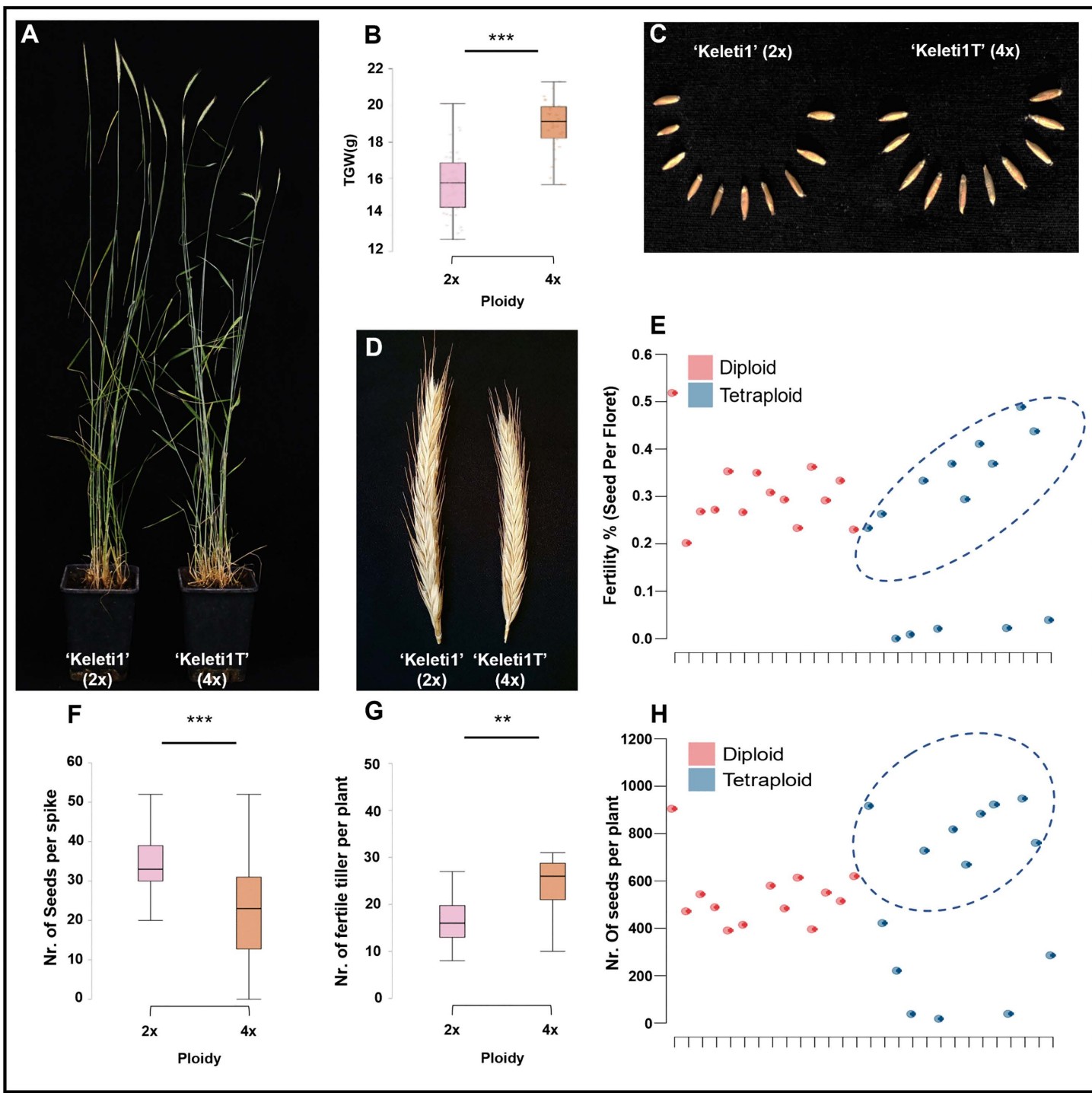

**Fig 2. Agronomic traits of diploid and tetraploid perennial rye. (A)** Representative images of fully developed diploid ('Keleti1', 2x) and tetraploid ('Keleti1T', 4x) plants. The image shows plants prior to the late harvest, which was preceded by an early harvest (S2 Fig), together resulting in the total number of fertile tillers per plant. **(B)** Boxplot showing thousand-grain weight (TGW) per plant. **(C)** Representative seed samples of 'Keleti1' (2x) and 'Keleti1T' (4x). **(D)** Representative spikes of diploid and tetraploid plants. **(E)** Reproductive efficiency expressed as seeds per floret for individual plants; ellipses indicate individuals with moderate to high values. **(F)** Boxplot showing the number of seeds per spike. **(G)** Boxplot showing the number of fertile tillers (spikes) per plant. Data include spikes from both early and late harvests, representing the total number of fertile tillers per plant. **(H)** Scatterplot

showing the total number of seeds per plant; dots represent individual plants, and ellipses indicate individuals with moderate to high values. Boxplots show medians, interquartile ranges, and whiskers extending to 1.5× the interquartile range. Statistical significance is indicated as (**: P < 0.01; ***: P < 0.001). Data underlying the presented analyses are available in S1 Table.

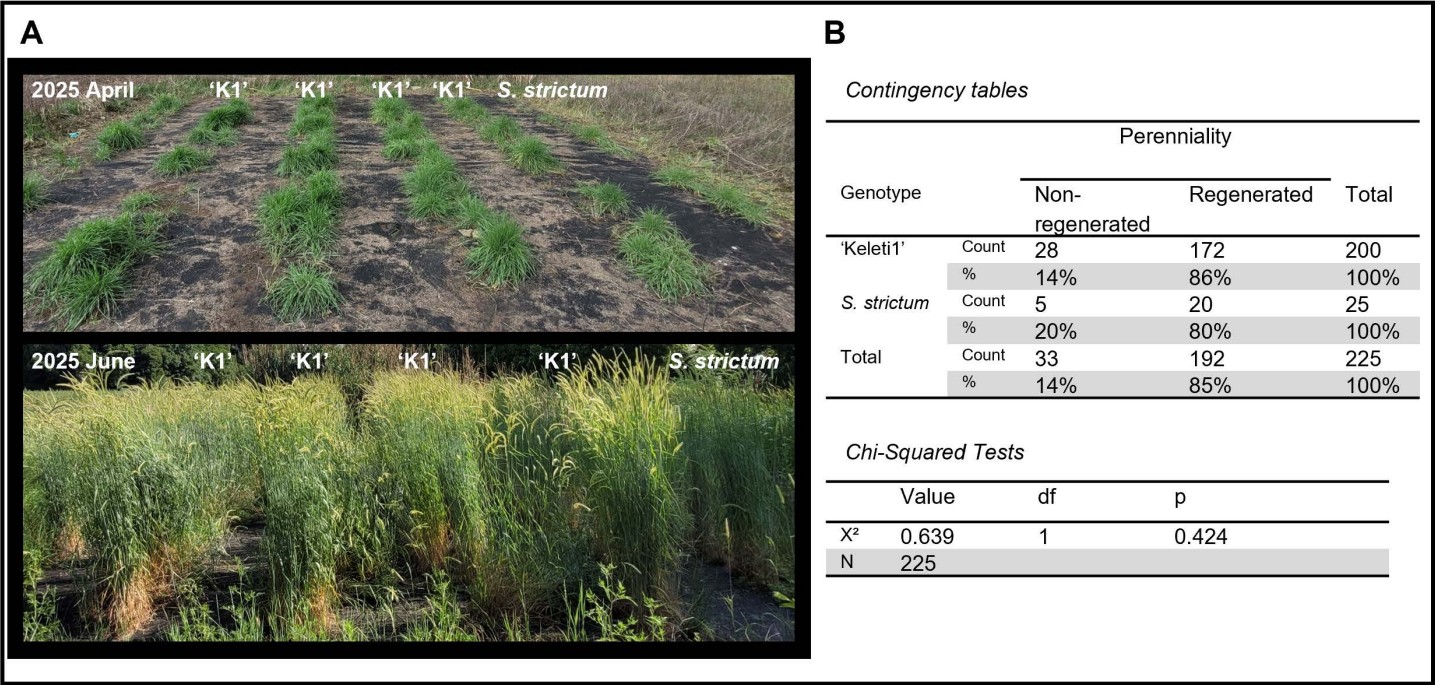

**A**

2025 April    'K1'    'K1'    'K1'  'K1'   *S. strictum*

2025 June    'K1'    'K1'    'K1'      'K1'    *S. strictum*

**B**

*Contingency tables*

| Genotype | | Perenniality | | |
|---|---|---|---|---|
| | | Non-regenerated | Regenerated | Total |
| 'Keleti1' | Count | 28 | 172 | 200 |
| | % | 14% | 86% | 100% |
| *S. strictum* | Count | 5 | 20 | 25 |
| | % | 20% | 80% | 100% |
| Total | Count | 33 | 192 | 225 |
| | % | 14% | 85% | 100% |

*Chi-Squared Tests*

| | Value | df | p |
|---|---|---|---|
| X² | 0.639 | 1 | 0.424 |
| N | 225 | | |

**Fig 3. Regrowth performance of diploid perennial rye compared with *Secale strictum*. (A)** Representative field images showing regrowth and development of the diploid perennial rye line 'Keleti1' and the perennial parent *S. strictum* during the second year following establishment. Images were taken in April and June of the 2025 growing season. **(B)** Statistical comparison of regrowth performance expressed as the proportion of plants showing successful regrowth. Pearson's chi-square test indicates that the regrowth percentage of diploid 'Keleti1' does not differ significantly from that of *S. strictum*.

## Cytogenetic differentiation of parental genomes in *S. cereale* and *S. strictum*

To establish reference karyotypes for parental genome identification, chromosomes of *S. cereale* cv. 'Kisvárdai' and *S. strictum* were analysed by fluorescence *in situ* hybridisation using the subtelomeric repeat pSc119.2 and a 5S rDNA probe. The combined probe set enabled unambiguous identification of all seven rye chromosomes in both species (Figs 4 and 5).

Within individual plants, FISH patterns were consistent across cells; however, heteromorphic chromosome pairs were frequently detected between homologues, particularly in *S. cereale*. Analysis of six *S. strictum* plants (n = 63 cells) revealed largely uniform karyotypes, whereas six *S. cereale* plants (n = 69 cells) displayed substantial intra- and inter-chromosomal polymorphism (S5, S6 Figs). Species-specific differences in pSc119.2 hybridisation were most prominent on chromosomes 1R and 6R, as well as on the long arms of chromosomes 3R, 4R, and 7R, and the short arm of chromosome 5R (Figs 4 and 5).

On chromosome 1R, *S. cereale* typically exhibited both telomeric and interstitial pSc119.2 signals on the short arm, whereas *S. strictum* lacked the interstitial short-arm signal and instead carried an additional interstitial site on the long

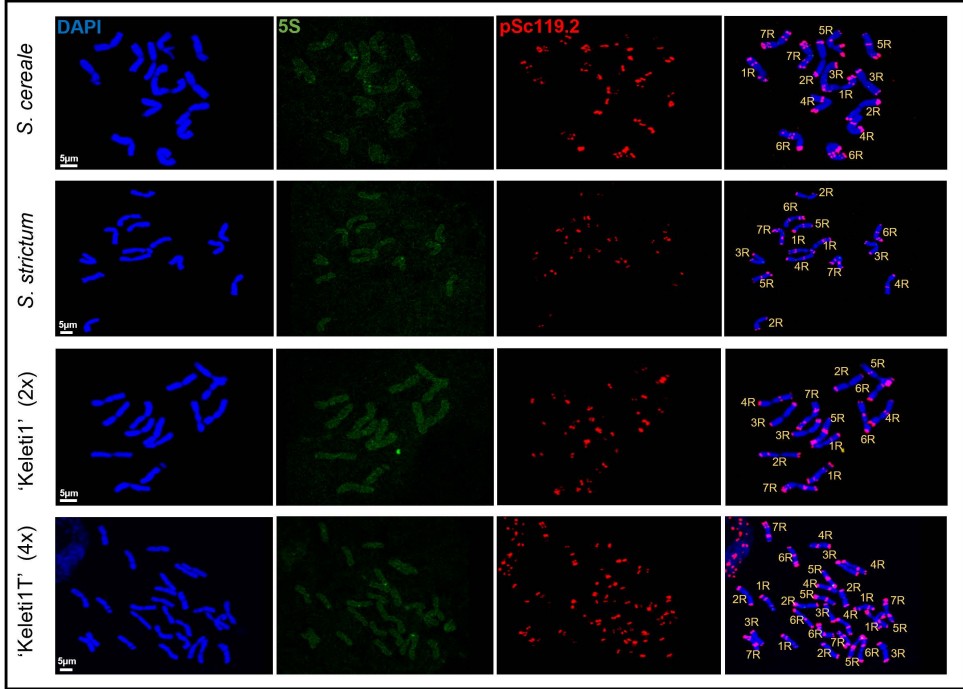

**Fig 4. Molecular cytogenetic identification of chromosomes in parental, diploid, and tetraploid rye lines.** Fluorescence *in situ* hybridisation (FISH) analysis of mitotic chromosomes from *Secale cereale*, *Secale strictum*, the diploid perennial line 'Keleti1' (2x), and its tetraploid derivative 'Keleti1T' (4x). Chromosomes were counterstained with DAPI (blue) and hybridised with 5S rDNA (green) and the subtelomeric repeat pSc119.2 (red). The rightmost column shows merged images with chromosome identification (1R–7R) based on signal distribution patterns. Scale bars = 5 µm.

arm (Figs 4 and 5; S6 Fig). Chromosome 2R showed the most conserved hybridisation pattern, with a single telomeric pSc119.2 signal on each arm in both species (Figs 4 and 5). Polymorphism was also detected on chromosome 3R, where reduced pSc119.2 signal intensity on one long-arm homologue occurred in a subset of *S. cereale* individuals (S5, S6 Fig). In addition, the position of the 5S rDNA locus differed between species, being pericentromeric on 3RL in *S. cereale* but centromeric in *S. strictum* (Figs 4 and 5). Chromosome 4R displayed conserved subtelomeric and interstitial pSc119.2 sites in both species; however, most *S. cereale* plants carried an additional weak telomeric signal on 4RL, providing a distinguishing feature from *S. strictum* (Figs 4 and 5; S6 Fig).

Differences were also evident on chromosome 5R, where the 5S rDNA locus was detected on both short arms in *S. cereale* but on only one short arm in *S. strictum* (Figs 4 and 5). Chromosome 6R showed extensive polymorphism in *S. cereale*, with variable numbers and positions of interstitial pSc119.2 signals on both arms, partially overlapping with the patterns observed in *S. strictum* (Figs 4 and 5; S6 Fig). Finally, chromosome 7R shared conserved short-arm pSc119.2 signals between species, whereas the long arm differed by the presence of an additional interstitial site in *S. strictum* (Figs 4 and 5).

Collectively, these species-specific and heteromorphic FISH patterns provided a robust cytogenetic framework for distinguishing parental chromosome segments and for tracking introgressed chromatin in hybrid derivatives.

## Parental chromosome composition of the diploid 'Keleti1'

FISH analysis of diploid 'Keleti1' revealed a mosaic chromosome composition consistent with its interspecific origin. Several chromosome arms could be assigned unambiguously to either *S. cereale* or *S. strictum* based on pSc119.2 and 5S rDNA signal patterns (Figs 4 and 5).

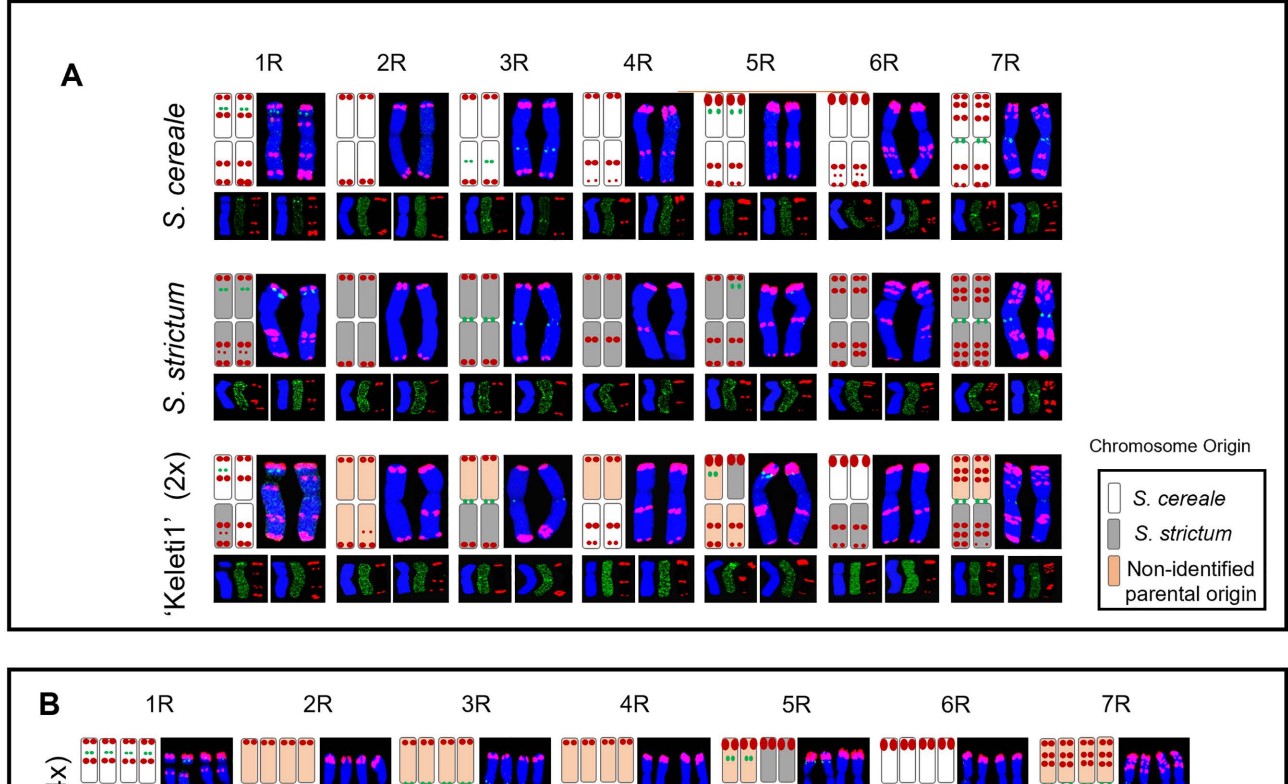

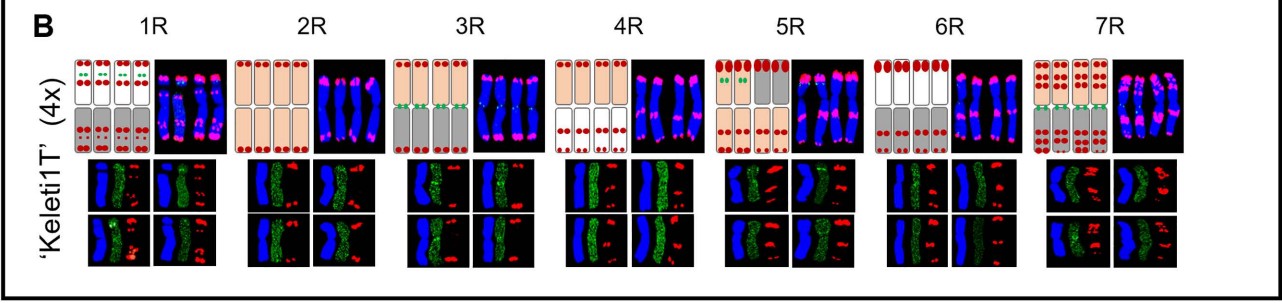

**Fig 5. Karyotypic layout and chromosome origin in parental, diploid, and tetraploid rye lines. (A)** Karyotypes of *Secale cereale*, *Secale strictum*, and the diploid perennial line 'Keleti1' (2x), arranged by homologous chromosome groups from 1R to 7R. For each chromosome group, representative homologous chromosomes are shown together with schematic summaries indicating inferred parental origin based on fluorescence *in situ* hybridisation (FISH) signal patterns. Chromosomes are shown with merged DAPI (blue), 5S rDNA (green), and pSc119.2 (red) signals. **(B)** Karyotype of the tetraploid derivative 'Keleti1T' (4x), with chromosomes arranged by homologous groups (1R–7R) and displayed as sets of four homologues. Representative merged images and corresponding schematic summaries of parental origin are shown for each chromosome group. Schematic colour coding indicates chromosome segments assigned to *S. cereale*, *S. strictum*, or regions of undetermined parental origin, as indicated in the key.

The 1RS chromosome arm of 'Keleti1' resembled that of *S. cereale* based on its pSc119.2 FISH pattern (n = 84; Figs 4 and 5). In three of the five plants analysed, the 1RL carried one interstitial and one telomeric pSc119.2 signal, matching the *S. cereale* pattern (S5, S6 Figs), whereas in the remaining two plants one of the 1RL homologues displayed two interstitial and one telomeric signal, characteristic of *S. strictum* (Figs 4 and 5). The 5S rDNA signal was detected at the subtelomeres of 1RS, although reduced signal intensity was observed on one homologue in a subset of plants (Figs 4 and 5). Chromosome 2R exhibited the conserved parental pattern, with telomeric pSc119.2 signals on both arms (Figs 4 and 5). However, an additional interstitial pSc119.2 site was detected on one long arm, a configuration not observed in either parent and likely arising from non-homologous recombination (Figs 4 and 5). The pSc119.2 hybridisation pattern of chromosome 3R was conserved across all genotypes, while the 5S rDNA locus in 'Keleti1' was located at the centromere,

consistent with *S. strictum* (Figs 4 and 5). Chromosome 4R showed conserved pSc119.2 signals on 4RS and an interstitial site on 4RL as observed in both parents (Figs 4 and 5); however, the telomeric signal on 4RL indicated a *S. cereale* origin and was more intense in 'Keleti1' than in the parental genotype (Figs 4 and 5). On chromosome 5R, the 5S rDNA locus was detected on only one of the homologous 5RS arms, while the remainder of the 5R karyotype was conserved (Figs 4 and 5). The 6RS of 'Keleti1' resembled *S. cereale*, whereas the 6RL corresponded to the *S. strictum* pattern (Figs 4 and 5). Finally, chromosome 7R displayed heteromorphism between homologues: one 7RL matched the *S. strictum* karyotype, while the other lacked the strong telomeric pSc119.2 signal, consistent with a terminal deletion (Figs 4 and 5).

## Cytogenetic changes following genome duplication in 'Keleti1T'

Cytological analysis of tetraploid plants confirmed their tetraploid status and revealed that the fluorescence *in situ* hybridisation patterns of chromosomes 3R, 5R and 7R were indistinguishable from those observed in the diploid progenitor 'Keleti1', indicating karyotypic conservation of these chromosomes following genome duplication (Figs 4 and 5). In contrast, chromosomes 1R, 2R, 4R and 6R exhibited limited polymorphism relative to the diploid line. For chromosome 1R, most tetraploid plants exhibited the same pSc119.2 hybridisation pattern as diploid 'Keleti1' (n = 73 cells; Figs 4 and 5). However, in one of the seven tetraploid individuals analysed, an alternative configuration was detected in which one 1RS homologue showed a reduced terminal pSc119.2 signal replaced by a weaker subterminal site (n = 11 cells; Figs 4 and 5; S6 Fig).

Chromosome 2R in all tetraploid plants corresponded to one of the diploids 'Keleti1' 2R variants, carrying a single pSc119.2 signal at each chromosome end (Figs 4 and 5). Similarly, six of the seven tetraploid lines displayed an identical 4RL pSc119.2 pattern to the diploid progenitor (n = 62 cells), whereas one individual lacked the telomeric pSc119.2 signal on both 4RL homologues (Figs 4 and 5; S5 Fig).

The FISH pattern of chromosome 6R was largely conserved between diploid and tetraploid plants; however, one tetraploid individual carried two heteromorphic 6R chromosomes, in which the short arm of one homologue contained an additional subtelomeric pSc119.2 band not observed in the diploid line (Figs 4 and 5; S5 Fig).

Despite the substantial variation in fertility observed among tetraploid plants, no clear association was detected between fertility and karyotype based on the *in situ* hybridisation markers used in this study. Individuals exhibiting identical FISH karyotypes displayed a wide range of fertility, from complete infertility to levels comparable to diploids (e.g., 0.36 seeds per floret).

## Discussion

Interspecific hybridisation combined with whole-genome duplication provides a route for introducing perennial growth habit and adaptive traits from wild relatives into cultivated cereals, but it also imposes substantial challenges related to genome organisation and reproductive stability. In rye, these challenges are accentuated by obligate outcrossing and extensive structural heterozygosity, which complicate both chromosome pairing and the retention of introgressed chromatin. In this study, we used the perennial interspecific rye derivative 'Keleti1' and its colchicine-induced tetraploid form 'Keleti1T' to examine how genome duplication affects agronomic traits and chromosome structure in a *Secale cereale* × *S. strictum* background.

### Agronomic effects of genome duplication in a perennial rye background

Genome duplication in 'Keleti1' resulted in a consistent reduction in plant height, an agronomically relevant trait due to its association with lodging resistance. The effect observed here contrasts with earlier reports in autotetraploid *S. cereale*, where plant height either increased or remained unchanged following genome duplication [17,33,34]. This discrepancy suggests that the phenotypic consequences of polyploidisation in rye are highly context-dependent and influenced by interspecific genome composition rather than ploidy level alone.

In major cereals such as wheat, reduced plant height has been a major contributor to yield improvement, primarily through the introgression of *Rht* dwarfing alleles during the Green Revolution [35,36]. In rye, the dominant dwarfing gene *Ddw1* confers reduced stature but is difficult to deploy effectively due to heterozygosity in open-pollinating populations [37–39]. Moreover, the performance of major dwarfing alleles in wheat is increasingly compromised under drought and heat stress [40,41]. In this context, the stature reduction observed following genome duplication in 'Keleti1T' suggests that polyploidisation may provide an alternative route to modifying plant architecture in rye without relying on single major-effect dwarfing genes.

Polyploidisation also increased thousand-grain weight, consistent with observations in autotetraploid rye and other polyploid cereals [17]. This increase occurred despite reduced fertility at the level of individual spikes, including fewer florets and lower seed set per floret. Importantly, these negative effects were compensated by an increased number of spikes per plant, resulting in no net reduction in total seed number per plant. Similar trade-offs between reproductive output per spike and kernel size have been reported previously in tetraploid rye [17], indicating that yield components respond differentially to genome duplication and that overall productivity depends on their balance rather than on individual traits alone.

Reduced fertility in perennial and interspecific rye derivatives has been reported repeatedly and is often associated with altered vernalisation requirements and meiotic irregularities [8,9,42]. While meiotic behaviour was not directly assessed in the present study, the variable seed set per floret in 'Keleti1T' suggests that genome duplication alone does not fully restore reproductive efficiency in this background. Nevertheless, the observed inter-individual variation suggests that this population could be explored in subsequent selection programmes.

## Cytogenetic consequences of interspecific hybridisation and polyploidisation

Using pSc119.2 and 5S rDNA probes, we established reference karyotypes for *S. cereale* and *S. strictum* and identified multiple species-specific and heteromorphic chromosome features. Consistent with earlier studies, chromosomes 1R, 6R, and the long arms of 3R, 4R, and 7R showed the greatest structural divergence between the two species [25,43]. These differences provided reliable cytogenetic markers for tracing parental chromosome segments in the hybrid derivatives.

The diploid 'Keleti1' line displayed a mosaic chromosome composition, with different parental origins detectable on opposite arms of the same chromosome and additional pSc119.2 sites absent from either parent. Similar reorganisation of repetitive sequences has been reported previously in perennial rye cultivars derived from *S. cereale* × *S. strictum* hybrids, including 'Kriszta' [43,44], and is consistent with frequent non-homologous recombination involving telomeric and subtelomeric regions [45].

At the tetraploid level, most chromosomes retained the FISH patterns observed in diploid 'Keleti1', indicating that genome duplication did not trigger widespread restructuring of repetitive sequences. Nevertheless, limited additional polymorphisms were detected on chromosomes 1R, 4R, and 6R in individual plants, demonstrating that structural heterozygosity persists following genome duplication. Similar intra-individual and inter-individual polymorphism of repetitive DNA–based cytogenetic markers has been reported in rye and in other cereals [46–49], including tetraploid wheat, highlighting the dynamic nature of repetitive sequences even within cytogenetically stable backgrounds [50]. The absence of a correlation between fertility and FISH-based karyotypes in this study suggests that fertility differences may be influenced by genomic rearrangements not detectable with the markers used, or by environmental factors such as the outcrossing nature of the species and potential limitations in tetraploid pollen availability [51].

## Implications for perennial rye improvement

The combined phenotypic and cytogenetic analyses presented here illustrate both the opportunities and limitations of using genome duplication to stabilise interspecific perennial rye derivatives. Polyploidisation modified key agronomic traits in a direction that may be favourable for breeding, particularly through increased grain size and reduced plant height,

while maintaining overall seed production. At the same time, structural heterozygosity and variable fertility remain significant constraints. A limitation of this study is the relatively small number of plants assessed for agronomic traits and karyotype analysis, which reduced the statistical power and the robustness of the comparative results. This limitation was due to the restricted number of successfully established tetraploid plants following colchicine treatment. Therefore, further studies involving larger populations and evaluations across subsequent generations are needed to strengthen the robustness of these comparisons.

Importantly, the cytogenetic framework established here provides practical tools for monitoring parental chromatin and structural variation during selection. The ability to distinguish *S. cereale* and *S. strictum* chromosome segments at the chromosome-arm level enables informed breeding decisions and facilitates the selection of lines with more stable karyotypes. Further work across successive generations will be required to determine how structural heterozygosity is maintained or resolved in tetraploid perennial rye and how this relates to fertility and long-term genome stability.

## Supporting information

**S1 Fig. Flow cytometric detection of mixoploidy in perennial rye.** Flow cytometry histogram showing relative nuclear DNA content of a mixoploid perennial rye plant. Distinct fluorescence peaks correspond to diploid (2x) and tetraploid (4x) nuclei; an additional minor peak corresponds to higher-ploidy nuclei (8x). The x-axis indicates relative fluorescence intensity (FL1), and the y-axis indicates event counts.
(TIF)

**S2 Fig. Representative diploid and tetraploid plants prior to early harvest.** Representative images of fully developed diploid ('Keleti1', 2x) and tetraploid ('Keleti1T', 4x) plants prior to the early harvest, which was followed by a late harvest (Fig. 2A) together resulting in the total number of fertile tillers per plant (Fig. 2G).
(TIF)

**S3 Fig. Additional agronomic traits of diploid and tetraploid perennial rye.** Boxplots comparing agronomic traits of diploid ('Keleti1', 2x) and tetraploid ('Keleti1T', 4x) plants: (A) Seed length (mm). (B) Seed width (mm). (C) Number of florets per spike. (D) Plant height (cm). (E) Spike length (cm). (F) Number of florets per plant. Boxplots show medians, interquartile ranges, and whiskers extending to 1.5× the interquartile range; points represent individual plants. Statistical significance is indicated as (***: $P < 0.001$; ns, not significant).
(TIF)

**S4 Fig. Regrowth of diploid and tetraploid perennial rye following harvest.** Representative images showing regrowth of diploid ('Keleti1', 2x) and tetraploid ('Keleti1T', 4x) perennial rye plants grown in large pots after spike harvest and tiller removal. Photographs were taken in July, August, and October 2024, illustrating progressive vegetative regrowth at both ploidy levels. Plants of both cytotypes (n = 14 each) exhibited comparable regrowth over time, with no visible differences in perennial capacity under the conditions tested.
(TIF)

**S5 Fig. Alternative fluorescence *in situ* hybridisation (FISH) signal patterns observed on mitotic chromosomes of *Secale cereale*, the diploid perennial line 'Keleti1' (2x), and its tetraploid derivative 'Keleti1T' (4x).** Chromosomes were counterstained with DAPI (blue). The second column shows hybridisation signals from the 5S rDNA probe (green), and the third column shows signals from the subtelomeric repeat pSc119.2 (red). The fourth column presents merged images with chromosome identification (1R–7R) based on signal distribution patterns. Scale bars = 5 µm.
(TIF)

**S6 Fig. Karyotypic layout and chromosome cutouts of rye chromosomes showing polymorphic signal patterns in parental, diploid, and tetraploid lines.** (A) Representative karyotypic layouts and chromosome cutouts of chromosomes 1R, 3R, 4R, and 6R from *Secale cereale* and the diploid perennial line 'Keleti1' (2x), arranged by homologous chromosome groups. (B) Karyotypic layout of the tetraploid derivative 'Keleti1T' (4x), with chromosomes 1R, 4R, and 6R displayed as sets of four homologues. For each chromosome group, individual chromosome cutouts are shown separately, including merged fluorescence *in situ* hybridisation (FISH) signals for DAPI (blue), 5S rDNA (green), and pSc119.2 (red), together with schematic summaries of signal distribution patterns. Scale bars = 5 µm.
(TIF)

**S1 Table. Morphological and Reproductive Traits of Diploid and Tetraploid Perennial rye ('Keleti1' and 'Keleti1T') Individual plant level measurements of growth and yield-related traits for diploid (2x; 'Keleti1') and tetraploid (4x; 'Keleti1T') perennial rye genotypes.** Missing data are indicated as "NA." Values represent direct measurements or derived means calculated per individual plant.
(XLSX)

## Acknowledgments

The authors thank the technical assistance of Mrs Ildiko Lakner Könyvesné and Mrs Barbara Péntek Krárné.

## Author contributions

**Conceptualization:** László Sági, Ákos Tarnawa, Dávid Polgari.

**Data curation:** Dávid Polgari.

**Formal analysis:** Ahmed Ali Hamad, Diána Makai, Ákos Tarnawa, Dávid Polgari.

**Funding acquisition:** László Sági, Adel Sepsi.

**Investigation:** Ahmed Ali Hamad, Diána Makai, Ákos Tarnawa, Adel Sepsi.

**Methodology:** Ahmed Ali Hamad, Adel Sepsi.

**Resources:** László Sági, Ákos Tarnawa.

**Supervision:** Adel Sepsi, Dávid Polgari.

**Validation:** László Sági, Dávid Polgari.

**Visualization:** Ahmed Ali Hamad, Diána Makai.

**Writing – original draft:** Ahmed Ali Hamad, Adel Sepsi, Dávid Polgari.

**Writing – review & editing:** Ahmed Ali Hamad, László Sági, Ákos Tarnawa, Adel Sepsi, Dávid Polgari.

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
