## [Decision Letter · Decision Letter 0]

11 Mar 2026

PONE-D-26-04864Molecular cytogenetic mapping of Secale strictum introgressions in a perennial tetraploid rye and its diploid progenitorPLOS One

Dear Dr. Sepsi,

Thank you for submitting your manuscript to PLOS ONE. After careful consideration, we feel that it has merit but does not fully meet PLOS ONE’s publication criteria as it currently stands. Therefore, we invite you to submit a revised version of the manuscript that addresses the points raised during the review process.

We look forward to receiving your revised manuscript.

Kind regards,

Aimin Zhang, Ph.D.

Academic Editor

PLOS One

**Journal Requirements:**

“The research was supported by the Hungarian Research, Development and Innovation Office (NKFIH, TKP2021-NKTA) (AS), the Hungarian National Laboratories Program (grant number RRF-2.3.1-21-2022-00007) (LS) and the Doctoral School of Plant Science of the Hungarian University of Agriculture and Life Science (AAH).”

5. Please note that funding information should not appear in any section or other areas of your manuscript. We will only publish funding information present in the Funding Statement section of the online submission form. Please remove any funding-related text from the manuscript.

6. We note that Figures 2 and 3 and Supplementary figure 3  in your submission contain copyrighted images. All PLOS content is published under the Creative Commons Attribution License (CC BY 4.0), which means that the manuscript, images, and Supporting Information files will be freely available online, and any third party is permitted to access, download, copy, distribute, and use these materials in any way, even commercially, with proper attribution. For more information, see our copyright guidelines: http://journals.plos.org/plosone/s/licenses-and-copyright.

1. You may seek permission from the original copyright holder of Figures 2 and 3 and Supplementary figure 3 to publish the content specifically under the CC BY 4.0 license.

Reviewers' comments:

Reviewer's Responses to Questions

**Comments to the Author**

1. Is the manuscript technically sound, and do the data support the conclusions?

Reviewer #1: Partly

2. Has the statistical analysis been performed appropriately and rigorously? 

Reviewer #1: Yes

3. Have the authors made all data underlying the findings in their manuscript fully available?

Reviewer #1: Yes

4. Is the manuscript presented in an intelligible fashion and written in standard English?

Reviewer #1: Yes

5. Review Comments to the Author

Reviewer #1: This manuscript needs minor revision as following:

1. In the comparison of agronomic traits and karyotype analysis between diploid and tetraploid plants, only 14 individuals were examined per cytotype. This relatively small sample size may affect the reliability of the comparative results. It is recommended that this limitation be explicitly acknowledged in the Discussion section.

2. There is an inconsistency between the phenotypic data and the representative plant images, raising a question about the use of terminology related to tillering. In the Results section, the statistical data indicate that the mean number of fertile tillers (spikes) per plant exceeds 17 in diploids and 24 in tetraploids. However, in the plant images shown in Figure 2a, the number of visible fertile tillers (effective tiller?) is considerably lower than these statistical values. This discrepancy suggests that the statistical data may be more representative of the total tiller rather than the fertile tiller. Is there a potential confusion between these two concepts?

3. In the evaluation of perennial growth habit, the diploid 'Keleti1' was compared in the field with its perennial parent S. strictum for regrowth rate (Figure 3). However, in the semi-outdoor pot experiment assessing the perennial habit of the tetraploid, it was only compared with the diploid 'Keleti1', lacking a direct comparison with the original perennial parent, S. strictum. While not a critical flaw, including this comparison would allow for a more comprehensive assessment of the impact of polyploidization on perennial growth habit.

4. Figure 3 compares the regrowth performance of the diploid 'Keleti1' and its parental species S. strictum. To enhance figure clarity, it is recommended that the two groups be clearly distinguished and labeled, facilitating direct visual comparison and interpretation by readers.

5. The Results section indicates substantial variation in seed number per plant among tetraploid individuals, with some reaching relatively high values. Concurrently, karyotypic differences (e.g., in pSc119.2 signal distribution) were observed among these tetraploid plants. It is therefore recommended to further analyze the cytological characteristics (e.g., specific FISH karyotypes) of these "high-value" tetraploid individuals to explore whether specific chromosomal karyotypes are associated with higher seed numbers. Correlating the phenotypic data with cytological data would enhance the logical coherence and scientific depth of the manuscript.

6. In the legends of Figure 2 and Supplementary Figure S2, it is recommended that significance levels be clearly indicated (e.g., **: P < 0.01; ***: P < 0.001) to improve result transparency and interpretability.

7. Please carefully check the formatting of the references. For instance, only the first word of article titles should be capitalized (see references 3, 6, 8, and 9, among others).

6. PLOS authors have the option to publish the peer review history of their article (what does this mean?). If published, this will include your full peer review and any attached files.

Reviewer #1: No

---

## [Author Response · Author response to Decision Letter 1]

23 Apr 2026

Reviewer 1 comments

Employing the newly developed 'Keleti1' line and its tetraploid derivative 'Keleti1T', this study examines the impact of genome duplication on agronomic traits and chromosomal architecture through integrated phenotypic and FISH-based cytogenetic analyses. The work is well-conceived and data-rich, providing important theoretical insights and practical cytogenetic tools for perennial rye improvement. The manuscript is of solid quality overall, though certain details and lines of reasoning could be further refined. The manuscript can be accepted after minor revision as the following comments:

1. In the comparison of agronomic traits and karyotype analysis between diploid and tetraploid plants, only 14 individuals were examined per cytotype. This relatively small sample size may affect the reliability of the comparative results. It is recommended that this limitation be explicitly acknowledged in the Discussion section.

RESPONSE

Thank you for highlighting this aspect of our study. We acknowledge that the sample size used in this study is limited, which we have now explicitly addressed in the Discussion section and we clarified the reasons of the experimental constraints.

Revised Text for the Discussion:

Page 22 Lines 487-493

‘A limitation of this study is the relatively small number of plants assessed for agronomic traits and karyotype analysis, which reduced the statistical power and the robustness of the comparative results. This limitation is due to the restricted number of successfully established tetraploid plants following colchicine treatment. Therefore, further studies involving larger populations and evaluations across subsequent generations are needed to strengthen the robustness of these comparisons.’

2. There is an inconsistency between the phenotypic data and the representative plant images, raising a question about the use of terminology related to tillering. In the Results section, the statistical data indicate that the mean number of fertile tillers (spikes) per plant exceeds 17 in diploids and 24 in tetraploids. However, in the plant images shown in Figure 2a, the number of visible fertile tillers (effective tiller?) is considerably lower than these statistical values. This discrepancy suggests that the statistical data may be more representative of the total tiller rather than the fertile tiller. Is there a potential confusion between these two concepts?

RESPONSE

Thank you for highlighting this issue. The values reported in the Results represent the mean number of fertile tillers (spikes) per plant across the entire reproductive season, during which mature spikes were collected in two rounds: an early harvest followed by a late harvest. The image presented in Fig. 2a shows plants prior to the late harvest and therefore reflects a lower number of spikes. Representative plants photographed prior to the first harvest are now included in the supplementary material (current S2 Figure) and show a higher number of spikes. This has now been explicitly clarified in the Materials and Methods and the corresponding figure legend, as follows:

Added text to the Materials and Methods:

P8 L145-150

‘Fertility and Phenotypic Assessment

Seed set per spike was calculated as the average number of grains per spike for each plant. The number of spikelets was recorded for each spike, and fertility was determined as the number of grains per spikelet. The number of fertile tillers was defined as the total number of spikes produced per plant during a single reproductive season (June-July), encompassing both the early harvest followed by a late harvest.’

Figure Legends:

P12 L261-263; P13 L268-269;

Fig. 2.

‘The image shows plants prior to the late harvest, which was preceded by an early harvest (Fig. S2), together resulting in the total number of fertile tillers per plant.’

and

‘Data include spikes from both early and late harvests, representing the total number of fertile tillers per plant.’

And P29 L670-675

‘S2 Figure Representative diploid and tetraploid plants prior to early harvest

Representative images of fully developed diploid (‘Keleti1’, 2x) and tetraploid (‘Keleti1T’, 4x) plants prior to the early harvest, which was followed by a late harvest (Fig. 2a) together resulting in the total number of fertile tillers per plant (Fig. 2g).’

3. In the evaluation of perennial growth habit, the diploid 'Keleti1' was compared in the field with its perennial parent S. strictum for regrowth rate (Figure 3). However, in the semi-outdoor pot experiment assessing the perennial habit of the tetraploid, it was only compared with the diploid 'Keleti1', lacking a direct comparison with the original perennial parent, S. strictum. While not a critical flaw, including this comparison would allow for a more comprehensive assessment of the impact of polyploidization on perennial growth habit.

RESPONSE

Thank you for this suggestion. The perennial growth habit of the diploid ‘Keleti1’ was confirmed to be comparable to that of its perennial parent, S. strictum, based on direct field comparisons of regrowth assessed in 2025 (Fig. 3). The aim of the semi-outdoor experiment was to assess whether perenniality is maintained or reduced following polyploidization by comparing the tetraploid with its diploid counterpart, and therefore only these genotypes were included. We acknowledge that growth of S. strictum plants alongside the ‘Keleti1’ and ‘Keleti1T’ genotypes in the semi-outdoor pot experiment would have enabled a more precise comparison, and we will consider including it as a parental control in future studies. Our semi-outdoor results indicate that perenniality can be effectively expressed in the polyploid state. Regrowth capacity and long-term persistence need to be further validated under controlled conditions and in field trials, which is one of our future motivation.

4. Figure 3 compares the regrowth performance of the diploid 'Keleti1' and its parental species S. strictum. To enhance figure clarity, it is recommended that the two groups be clearly distinguished and labeled, facilitating direct visual comparison and interpretation by readers.

RESPONSE

Thank you for this important observation, we have now labelled the figure to enhance its comprehensibility.

5. The Results section indicates substantial variation in seed number per plant among tetraploid individuals, with some reaching relatively high values. Concurrently, karyotypic differences (e.g., in pSc119.2 signal distribution) were observed among these tetraploid plants. It is therefore recommended to further analyze the cytological characteristics (e.g., specific FISH karyotypes) of these "high-value" tetraploid individuals to explore whether specific chromosomal karyotypes are associated with higher seed numbers. Correlating the phenotypic data with cytological data would enhance the logical coherence and scientific depth of the manuscript.

RESPONSE

Whilst fertility varied widely among the tetraploid plants, we can confirm that no correlation was observed between fertility and the karyotype distribution of the in situ hybridisation probes used in this study. Plants exhibiting identical karyotypes (based on the applied FISH probe)s ranged from complete infertility to fertility levels comparable to diploids. These findings suggest that additional karyotypic differences not detectable with the current methodology, or environmental factors, may contribute to the observed variation in fertility.

We added the following sentence to the Results section:

P18 L405-409

‘Despite the substantial variation in fertility observed among tetraploid plants, no clear association was detected between fertility and karyotype based on the in situ hybridisation markers used in this study. Individuals exhibiting identical FISH karyotypes displayed a wide range of fertility, from complete infertility to levels comparable to diploids (e.g., 0.36 seeds per floret).’

And the following sentence is included in the Discussion:

P21, L476-480

‘The absence of a correlation between fertility and FISH-based karyotypes in this study suggests that fertility differences may be influenced by genomic rearrangements not detectable with the markers used, or by environmental factors such as the outcrossing nature of the species and potential limitations in tetraploid pollen availability (Pfahler et al., 1987).’

6. In the legends of Figure 2 and Supplementary Figure S2, it is recommended that significance levels be clearly indicated (e.g., **: P < 0.01; ***: P < 0.001) to improve result transparency and interpretability.

RESPONSE

Significance levels are now indicated in the figure legends.

7. Please carefully check the formatting of the references. For instance, only the first word of article titles should be capitalized (see references 3, 6, 8, and 9, among others).

RESPONSE

Thank you for highlighting the formatting errors, we have corrected the reference list according to the Journal’s specifications.

---

## [Decision Letter · Decision Letter 1]

27 Apr 2026

Molecular cytogenetic mapping of Secale strictum introgressions in a perennial tetraploid rye and its diploid progenitor

PONE-D-26-04864R1

Dear Dr. Sepsi,

We’re pleased to inform you that your manuscript has been judged scientifically suitable for publication and will be formally accepted for publication once it meets all outstanding technical requirements.

Kind regards,

Aimin Zhang, Ph.D.

Academic Editor

PLOS One

Additional Editor Comments (optional):

Reviewers' comments:

Reviewer's Responses to Questions

**Comments to the Author**

1. If the authors have adequately addressed your comments raised in a previous round of review and you feel that this manuscript is now acceptable for publication, you may indicate that here to bypass the “Comments to the Author” section, enter your conflict of interest statement in the “Confidential to Editor” section, and submit your "Accept" recommendation.

Reviewer #1: All comments have been addressed

2. Is the manuscript technically sound, and do the data support the conclusions?

Reviewer #1: (No Response)

3. Has the statistical analysis been performed appropriately and rigorously? 

Reviewer #1: (No Response)

4. Have the authors made all data underlying the findings in their manuscript fully available?

Reviewer #1: (No Response)

5. Is the manuscript presented in an intelligible fashion and written in standard English?

Reviewer #1: (No Response)

6. Review Comments to the Author

Reviewer #1: Regarding the issues I raised, the authors have either substantially revised and improved the manuscript or provided reasonable explanations.

7. PLOS authors have the option to publish the peer review history of their article (what does this mean?). If published, this will include your full peer review and any attached files.

Reviewer #1: No

---

## [Editor Report · Acceptance letter]

PONE-D-26-04864R1

PLOS One

Dear Dr. Sepsi,

I'm pleased to inform you that your manuscript has been deemed suitable for publication in PLOS One. Congratulations! Your manuscript is now being handed over to our production team.

Kind regards,

on behalf of

Prof. Aimin Zhang

Academic Editor

PLOS One